# Diversity-decomposition relationships in forests worldwide

**Liang Kou[1,2†]\*, Lei Jiang[1,2†], Stephan Hättenschwiler[3], Miaomiao Zhang[4], Shuli Niu[1,2], Xiaoli Fu[1,2], Xiaoqin Dai[1,2], Han Yan[1,2], Shenggong Li[1,2]\*, Huimin Wang[1,2]\***

[1]Qianyanzhou Ecological Research Station, Key Laboratory of Ecosystem Network Observation and Modeling, Institute of Geographic Sciences and Natural Resources Research, Chinese Academy of Sciences, Beijing, China; [2]College of Resources and Environment, University of Chinese Academy of Sciences, Beijing, China; [3]CEFE, Univ. Montpellier, CNRS, EPHE, IRD, Univ. Paul-Valéry, Montpellier, France; [4]State Key Laboratory of Tree Genetics and Breeding, Key Laboratory of Tree Breeding and Cultivation of State Forestry Administration, Research Institute of Forestry, Chinese Academy of Forestry, Beijing, China

**Abstract** Plant species diversity affects carbon and nutrient cycling during litter decomposition, yet the generality of the direction of this effect and its magnitude remains uncertain. With a meta-analysis including 65 field studies across the Earth's major forest ecosystems, we show here that decomposition was faster when litter was composed of more than one species. These positive biodiversity effects were mostly driven by temperate forests but were more variable in other forests. Litter mixture effects emerged most strongly in early decomposition stages and were related to divergence in litter quality. Litter diversity also accelerated nitrogen, but not phosphorus release, potentially indicating a decoupling of nitrogen and phosphorus cycling and perhaps a shift in ecosystem nutrient limitation with changing biodiversity. Our findings demonstrate the importance of litter diversity effects for carbon and nutrient dynamics during decomposition, and show how these effects vary with litter traits, decomposer complexity and forest characteristics.

**\*For correspondence:**
koul@igsnrr.ac.cn (LK);
lisg@igsnrr.ac.cn (SL);
wanghm@igsnrr.ac.cn (HW)

†These authors contributed equally to this work

## Introduction

Forests mediate biosphere-atmosphere carbon (C) dynamics via primary productivity (*Huang et al., 2018*; *Liang et al., 2016*) and decomposition (*Handa et al., 2014*; *Hooper et al., 2012*). The biodiversity of organisms involved in C and nutrient cycling can modify these ecosystem processes, for example, during the decomposition of plant litter that typically occurs in mixtures (*Chomel et al., 2016*; *Hättenschwiler et al., 2005*). Litter mixtures can decompose at different rates than would be predicted from the rates of the individual component species, resulting in non-additive effects with either faster (synergistic effects) or slower (antagonistic effects) decomposition (*Gartner and Cardon, 2004*; *Wardle et al., 1997*). Synergistic effects may result from fungi-driven nitrogen (N) transfer among different litter types (*Lummer et al., 2012*; *Schimel and Hättenschwiler, 2007*) or through complementary resource use among microbial decomposers or detritivores (*Gessner et al., 2010*; *Vos et al., 2013*). On the other hand, antagonistic effects may result from inhibiting or recalcitrant compounds (e.g. lignin and polyphenols) present in one litter type that negatively affect the decomposition of the whole mixture of litter (*Hättenschwiler et al., 2005*). Despite the accumulating number of case studies, it remains difficult to generalize these data beyond the specific context of the different studies, hindering a more general understanding of the importance of litter diversity for biogeochemical cycling in forests.

The magnitude and direction of mixing effects may depend on several aspects of litter diversity, related to species number (richness), combinations of species or plant functional types (composition), relative abundance (evenness), and functional dissimilarity (spread in litter trait space) of litter species within mixtures (*Chapman and Koch, 2007*; *Otsing et al., 2018*). Moreover, other plants from different life forms, such as understory shrubs and herbs, contribute to the litter pool of forests, but are only rarely taken into account in litter mixture studies (*Chomel et al., 2016*). Another aspect of the world's forests not commonly considered, is that the majority of forests are heavily managed and tree species composition does not always represent the community that would naturally establish without management. Planting of productive, but exotic species, in particular may considerably reduce species richness and disrupt the relationships between tree-derived organic matter and decomposer communities, which have been described as the so-called 'home field advantage' (*Gholz et al., 2000*; *Veen et al., 2015*). Accordingly, litter diversity effects on decomposition may differ between naturally established forests and planted forests. All these different aspects are not commonly addressed together in the different studies, which also differ widely in the total number of species included, the duration of the experiment and in other facets of the experimental context. All these sources of variability among individual studies make it difficult to generalize the findings and to detect general patterns of litter diversity effects on decomposition.

The fact that the process of decomposition itself is constantly changing over time is another challenge in the understanding of biodiversity effects on litter decomposition. The largest part of labile C and nutrients available in leaf litters may be consumed by decomposers or leached with rainfall early in the decomposition process (*Gessner et al., 2010*), which potentially reduces the differences in litter chemistry among litter types during decomposition. For instance, previous studies reported that synergistic effects dominated early-stage decomposition, which disappeared (*Santonja et al., 2019*) or switched to antagonistic effects (*Butenschoen et al., 2014*) in later stages of decomposition. Furthermore, mixing effects may differ among distinct biomes characterized by different environmental conditions, affecting not only the composition of plant communities and the litter they produce, but also the structure of decomposer communities and their physiological constraints (*Fierer et al., 2009*; *Fierer et al., 2012*). In high-latitude forest biomes, lower temperatures may limit microbial activity and growth (*Davidson and Janssens, 2006*; *Zhou et al., 2016*), consequently affecting the biodiversity-dependence of decomposition. Consequently, non-additive effects may be more prominent in warmer climates, because lower temperature limitation of decomposers may accentuate their dependency on substrate quality and diversity (*Duan et al., 2013*). These spatio-temporal variations call for a more general understanding of litter diversity effects and how they may vary among the Earth's major forests.

Towards this goal, we assembled a global dataset encompassing 284 plant species combinations from 65 field studies on decomposition of leaf litter mixtures in five forest biomes (*Figure 1*; *Supplementary file 1*). We focused on three key decomposition variables associated with C and nutrient cycling (mass loss, N and phosphorus (P) release), and we paid particular attention to how mixing effects may shift during decomposition. We considered richness, evenness, and composition (deciduous vs. evergreen plant functional types) of litter mixtures, as well as forest types (natural vs. planted forests) and the inclusion or exclusion of understory species as driving factors. To assess how mixture effects may depend on environmental context and chemical characteristics of the litter mixtures, we included site-specific climatic variables and initial chemical trait divergence among component leaf litters within mixtures as additional variables. Specifically, we aimed to address the following questions: (1) Are mixture effects on mass loss, N and P release generally non-additive at the global scale, and do they differ among forest biomes? (2) Do mixture effects change during the process of decomposition (i.e. are they stronger in early vs. late decomposition stages)? (3) Are mixture effects related to the divergence of specific litter traits among constituent species? (4) Do mixture effects differ between natural and planted forests?

## Results

Across biomes, the average mass loss and N release measured at final harvest (i.e. excluding data from intermediate harvests in the studies with several harvest dates) were higher in litter mixtures compared to those expected from single species litter, but there was no effect on P release (*Figure 2*). The reported mixture effects varied among biomes and depended on the stage of

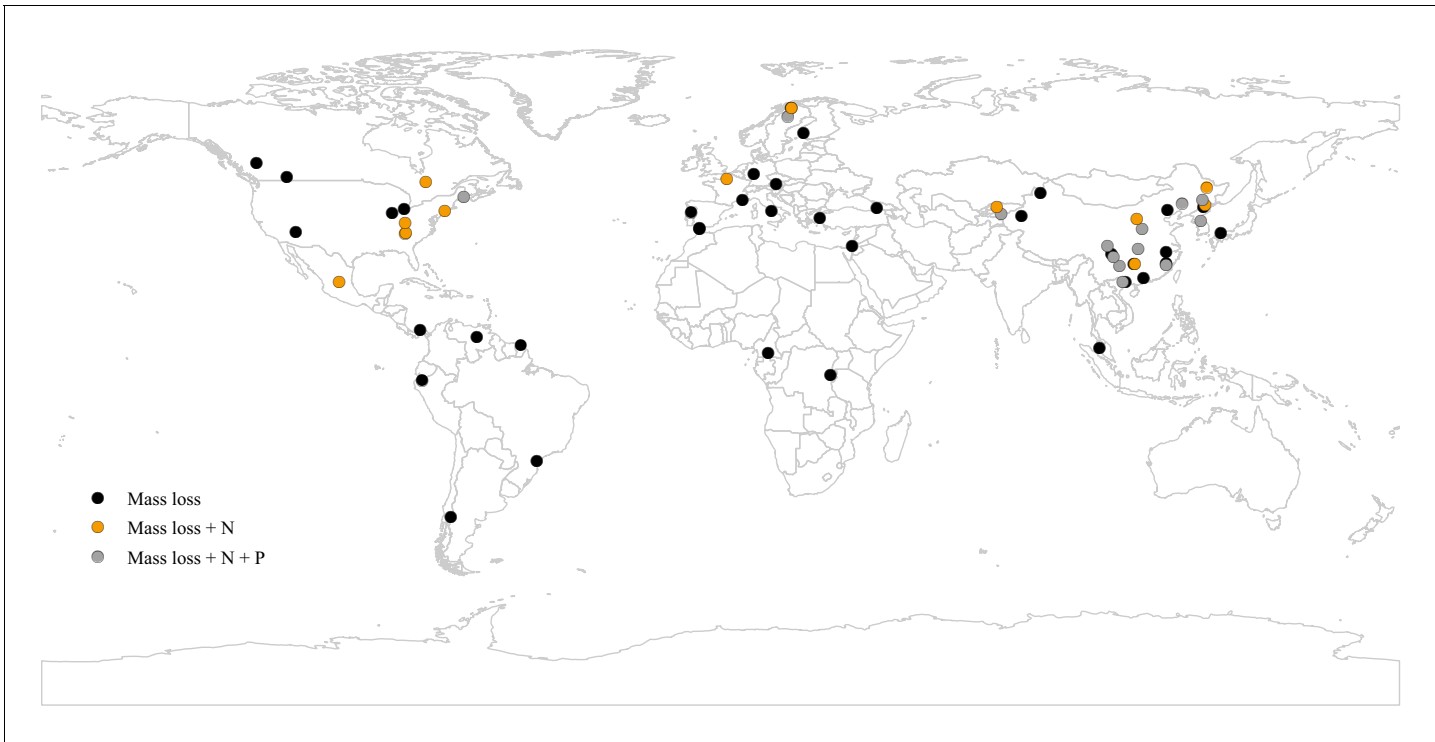

**Figure 1.** A map showing the geographical distribution of the 63 study sites included in the meta-analysis. Black circles denote all studies where only mass loss was assessed, yellow circles denote the studies where also N release was measured, and grey circles denote the studies where mass loss, N release, and P release were measured.

decomposition. Except for boreal forests, the effect size for mass loss was generally positive but was only significantly so in temperate and subtropical forests. The effect size for N release was significantly positive for temperate forests, showed a positive trend for boreal forests and a negative trend for subtropical forests (there was only one observation in each of tropical and Mediterranean forests). The effect sizes for P release were close to or below zero with high variability and low sample sizes. Regarding 10% mass loss intervals, the synergistic effects on mass loss were particularly clear in the range from 10% to 40% of mass loss (effect sizes: 7% to 10%), and decreased sharply in the range from 40% to 60% of mass loss (effect sizes: 2% to 3%) and further decreased gradually in later decomposition stages beyond 60% mass loss (*Figure 3*). At mass losses below 10% and above 60%, litter mixing had purely additive effects on mass loss. The dynamics of mixture effects on N release during the decomposition process were more complex than for mass loss. We found antagonistic effects on N release at 0–10% mass loss, additive effects during the three intervals from 10–20%, 40–50% and 60–70% mass loss, and synergistic effects during the four intervals from 20–30%, 30–40%, 50–60% and 70–100% (*Figure 3*). The litter mixing effects on P release were consistently additive throughout the process of decomposition (*Figure 3*).

To explore the striking differences in mixture effects on nutrient dynamics in more detail, we compared N and P losses during decomposition across all litter mixtures with those across all single species litter. Relative to the initial amount of N contained in litter, N release rates from mixtures (35.4% on average) were about twice as high compared to those from single species litter (18.7% on average, p=0.042; *Figure 2—figure supplement 1A*). In contrast, P release rates from mixtures (33.2% on average) were not significantly different compared to those from single species litter (29.7% on average; *Figure 2—figure supplement 1A*). The same patterns were observed for the absolute release rates of N and P (*Figure 2—figure supplement 1B,C*). The absolute release rate of N was higher in litter mixtures compared to single species litter (4.35 mg g$^{-1}$ vs. 2.51 mg g$^{-1}$; *Figure 2—figure supplement 1B*), while that of P was about the same in litter mixtures and single species litter (0.16 mg g$^{-1}$ vs. 0.13 mg g$^{-1}$; *Figure 2—figure supplement 1C*).

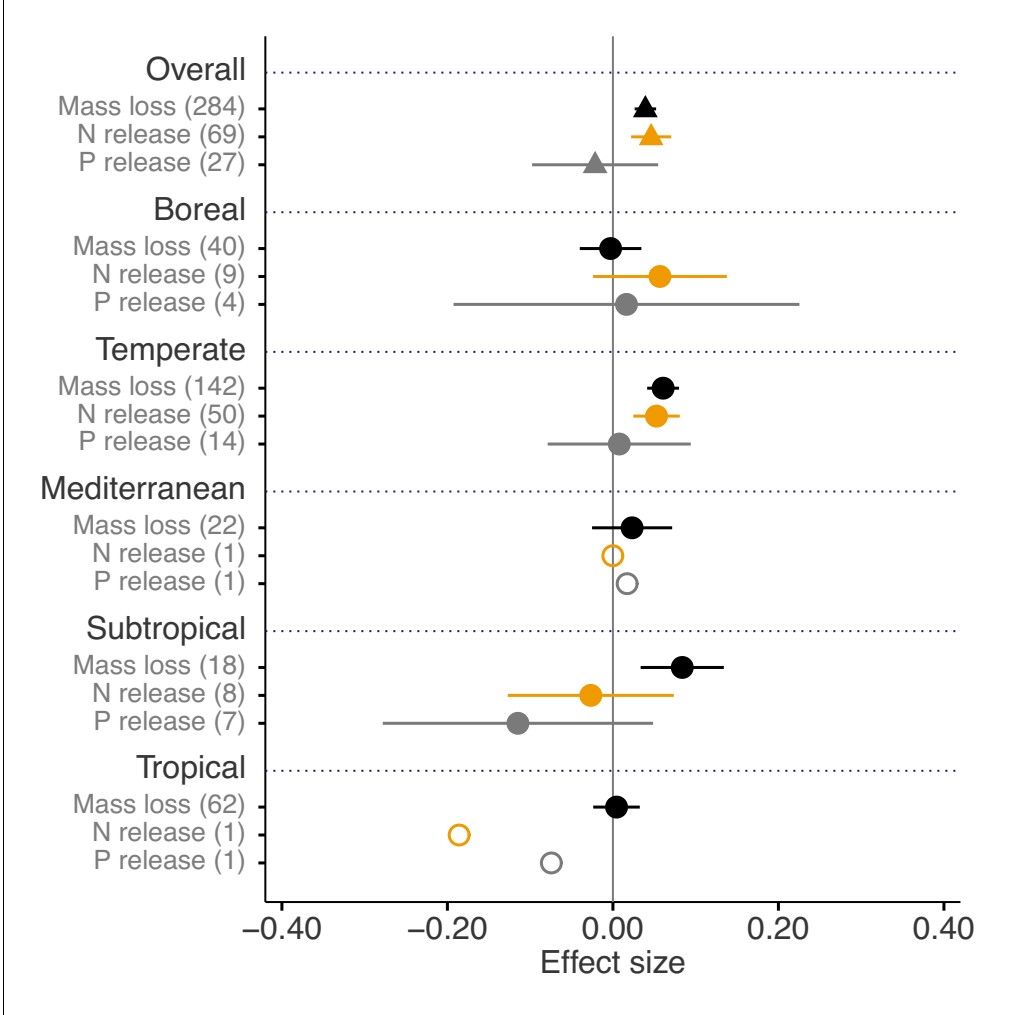

**Figure 2.** Litter mixture effects on mass loss (solid black triangles or circles), N release (solid yellow triangles or circles), and P release (solid grey triangles or circles) at the global scale (overall average, triangles) and for each of the five different biomes separately (circles). Effect sizes were calculated as the log response ratio between observed and expected values for each response variable and 95% bootstrap confidence intervals (CI) are shown. Data from the final harvest time of the decomposition experiment (if there was more than one harvest) were used to calculate the effect size of each variable. The solid vertical black line indicates no effects (mean effect size = 0, observed values = expected values). Mean effect size >0 indicates synergistic effects (observed values > expected values), while mean effect size <0 indicates antagonistic effects (observed values < expected values). The non-additive effects (synergistic or antagonistic) are significant at $\alpha$ = 0.05 if the CI of the effect size does not overlap 0, while the effects are additive when there is overlap. The values in parentheses denote the number of observations. The open circles indicate sample sizes that were too low to run statistic.

The online version of this article includes the following figure supplement(s) for figure 2:

**Figure supplement 1.** Nitrogen and P release from single litter species treatments and litter mixtures across all studies included in our meta-analysis.

**Figure supplement 2.** Overall average of N and P release and for each species richness level separately.

The mixture effects on litter mass loss were further influenced by the richness, plant functional type composition (deciduous and evergreen foliage), and evenness of litter mixtures, as well as forest stand structure and mesh size of litterbags (*Figures 4* and *5*). Specifically, synergistic effects emerged when litter mixtures were composed of two (n = 196) or three (n = 41) species while additive effects predominated when four (n = 34) or six species (n = 13) were included in a mixture. The synergistic effects were present regardless of the ratio (mass-based) of component litter species in a mixture, i.e. equal (n = 225) or non-equal ratios (n = 59). Moreover, synergistic effects were always

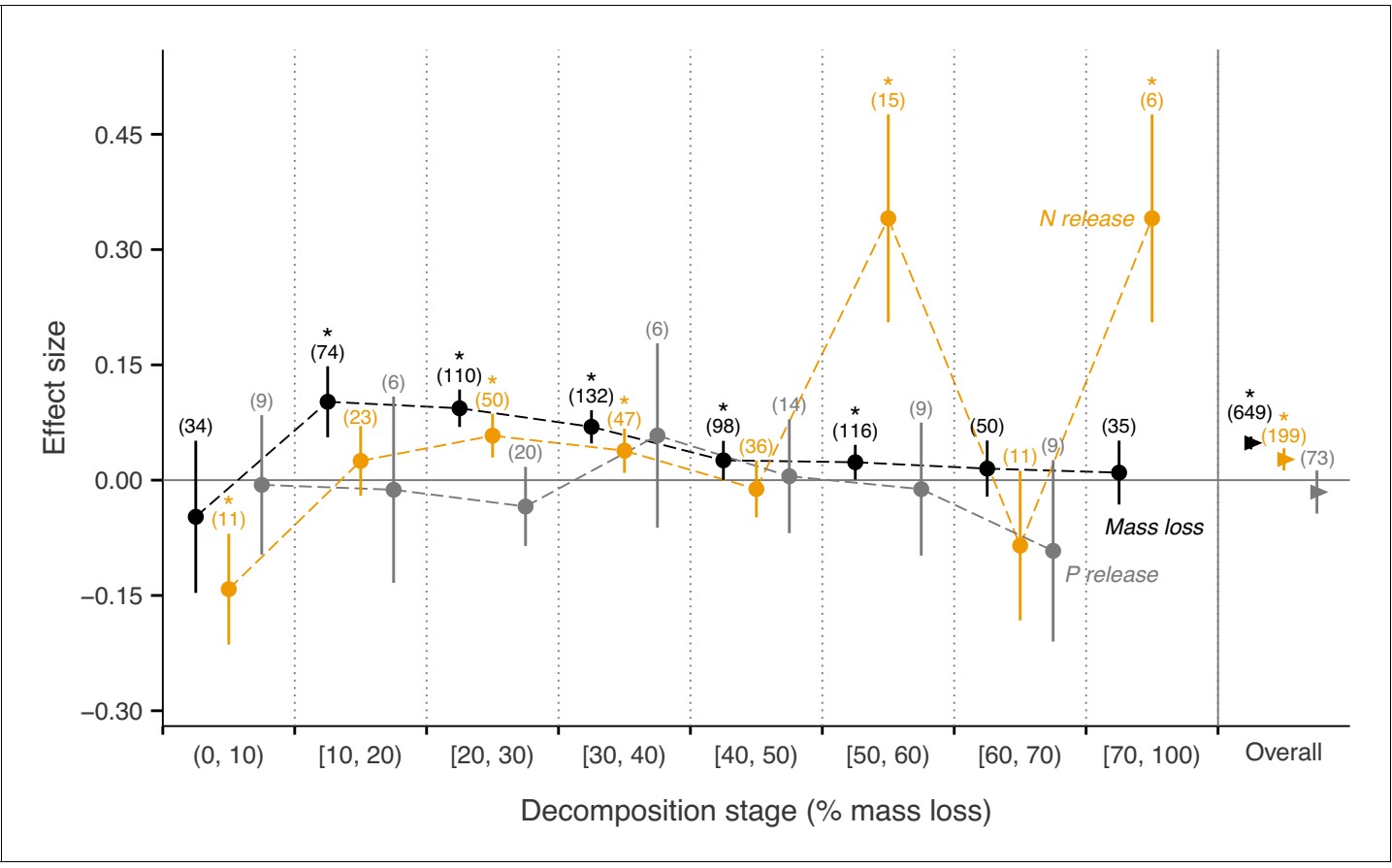

**Figure 3.** Litter mixture effects on mass loss (solid black triangles or circles), N release (solid yellow triangles or circles), and P release (solid grey triangles or circles) at different mass-loss intervals during the decomposition process (% mass loss indicated the expected values calculated from the mass loss of the individual component species). Effect sizes were calculated as the log response ratio between observed and expected values for each response variable and 95% bootstrap confidence intervals (CI) are shown. Data at each harvest time of the decomposition experiment were used to calculate the effect size of each variable. We created intervals of 10% increments in mass loss up to 70% of mass loss, but did not distinguish further the final interval from 70% to 100%, because of only few observations within this range of mass loss. The slightly different positions of symbols within each interval is to better differentiate among curves of mass loss, N release, and P release and has no other meaning. The solid horizontal black line indicates no effects (mean effect size = 0, observed values = expected values). Mean effect size >0 indicates synergistic effects (observed values > expected values), while mean effect size <0 indicates antagonistic effects (observed values < expected values). The non-additive effects (synergistic or antagonistic) are significant at $\alpha$ = 0.05 if the CI of the effect size does not overlap with the zero line, while the effects are additive when there is overlap. Values in parentheses above the symbols denote the number of observations and asterisks indicate a significant mixture effect (i.e. a significant deviation from 0).

found when litters from deciduous species were in the mixtures, irrespective of whether or not they were combined with evergreen species. On the contrary, when only evergreen species were mixed together, the effects remained additive. Forest stand structure mediated the species diversity effects that also depended on the type of forests. In natural forests, the synergistic effects emerged for litter mixtures composed of the canopy tree species but not for other combinations of litter mixtures when understory species were also included. However, we found the opposite pattern in planted forests (*Figure 4*). Litter mesh size also had an impact on the diversity effect. For mesh sizes between 1 mm and 2 mm, there was a clear synergistic effect (*n* = 148), while smaller mesh sizes (<1 mm, *n* = 72) and larger mesh sizes (>2 mm, *n* = 109) resulted in additive effects (*Figure 5*).

The effect sizes for mass loss were negatively correlated with MAP ($r^2$ = 0.049, p<0.001; *Figure 6B*), but they showed no correlation with MAT (p>0.05; *Figure 6A*). Differences in several litter quality parameters between component litters in mixtures had an apparent impact on the effect sizes in mass loss. During the period of decomposition with the strongest non-additive mixing effects (10–40% mass loss), the effect sizes for mass loss were negatively correlated with the divergence in

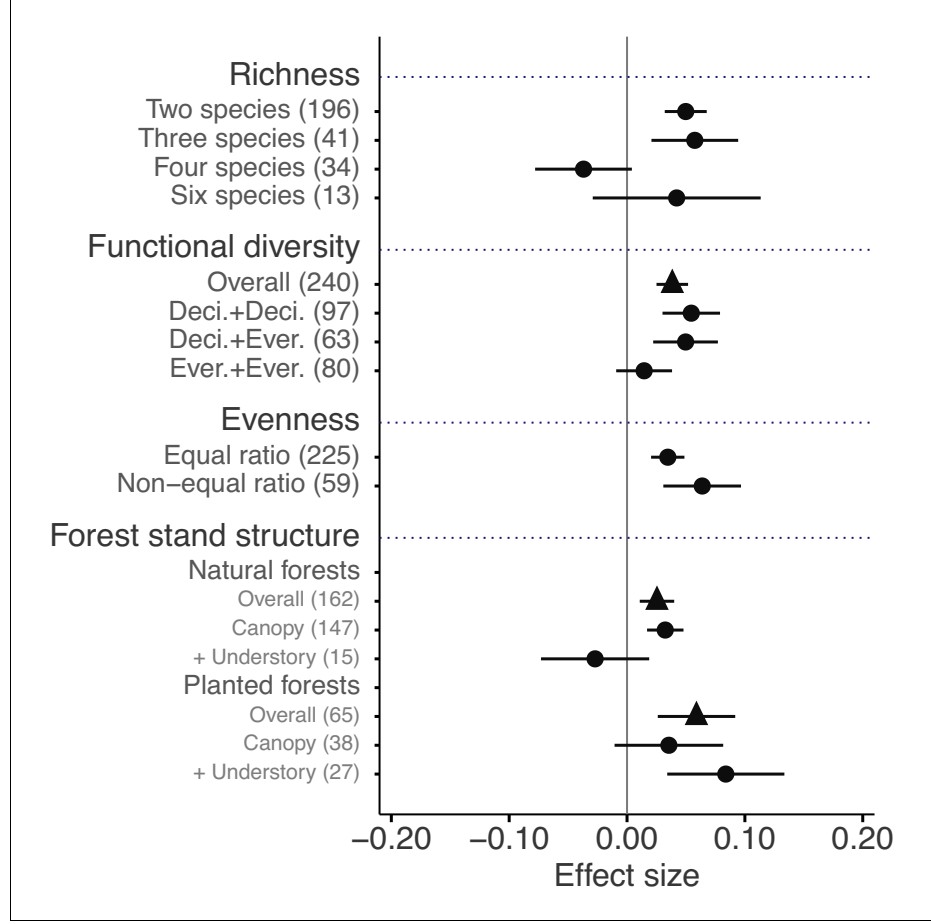

**Figure 4.** Litter mixture effects on mass loss as a function of the number of species present in litter mixtures, functional diversity (here defined on the basis of leaf habit), evenness in mixtures, and forest stand structure, which is distinguished between natural and planted forests. Effect sizes were calculated as the log response ratio between observed and expected values for each response variable and 95% bootstrap confidence intervals (CI) are shown. When there were several harvests through time, only the data from the final harvest were used for the calculation of effect sizes. The solid vertical black line indicates no effects (mean effect size = 0, observed values = expected values). Mean effect size >0 indicates synergistic effects (observed values > expected values), while mean effect size <0 indicates antagonistic effects (observed values < expected values). The non-additive effects (synergistic or antagonistic) are significant at $\alpha$ = 0.05 if the CI of the effect size does not overlap 0, while the effects are additive when there is overlap. The values in parentheses denote the number of observations. Deci. and Ever. stand for deciduous and evergreen woody species, respectively. Forest stand structure distinguishes between the overall effect (triangles), the effect of exclusively canopy species ('canopy'), and the effect when understory plant species were included in the studies ('+ understory').

The online version of this article includes the following figure supplement(s) for figure 4:

**Figure supplement 1.** Litter mixture effects on mass loss for each biome separately and distinguishing among overall and species richness level-specific effects.

**Figure supplement 2.** Initial quality of leaf litter from deciduous and evergreen species included in the meta-analysis.

the initial N:P ratio of the litter species included in the mixtures ($r^2$ = 0.076, p<0.05; *Figure 7I*). In contrast, there were positive correlations between the effect size of litter mass loss and the divergence in initial P ($r^2$ = 0.246, p<0.001; *Figure 7B*), K ($r^2$ = 0.445, p<0.001; *Figure 7C*), Ca ($r^2$ = 0.360, p<0.001; *Figure 7D*), Mg ($r^2$ = 0.346, p<0.001; *Figure 7E*), cellulose ($r^2$ = 0.388, p<0.001; *Figure 7G*), and lignin:N ratio ($r^2$ = 0.335, p<0.001; *Figure 7J*) among litter species present in the mixtures.

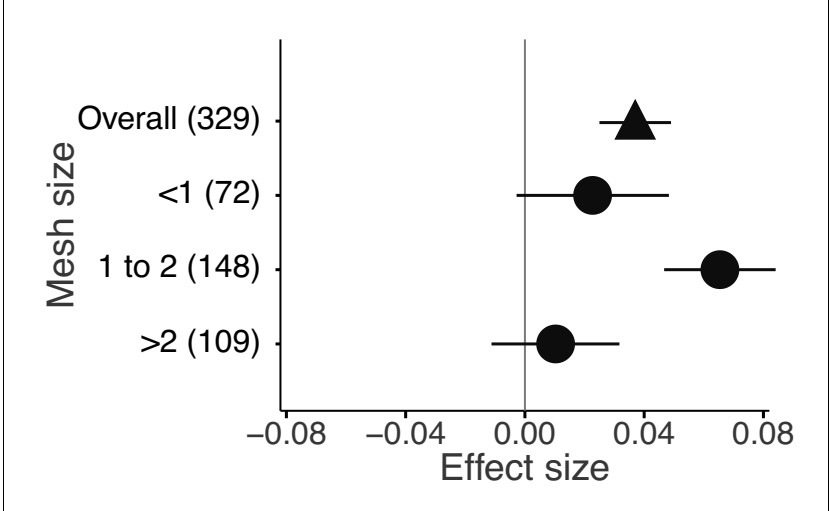

**Figure 5.** Litter mixture effects on mass loss from litterbags with different mesh sizes. Effect sizes were calculated as the log response ratio between observed and expected values for each response variable and 95% bootstrap confidence intervals (CI) are shown. Data from the final harvest of the decomposition experiment were used to calculate the effect sizes. The solid vertical black line indicates no effects (mean effect size = 0, observed values = expected values). Mean effect size >0 indicates synergistic effects (observed values > expected values), while mean effect size <0 indicates antagonistic effects (observed values < expected values). The non-additive effects (synergistic or antagonistic) are significant at α = 0.05 if the CI of the effect size does not overlap 0, while the effects are additive when there is overlap. The values in parentheses denote the number of observations. The online version of this article includes the following figure supplement(s) for figure 5:

**Figure supplement 1.** Litter mixture effects on mass loss from litterbags for each biome separately and distinguishing among overall and mesh size-specific effects.

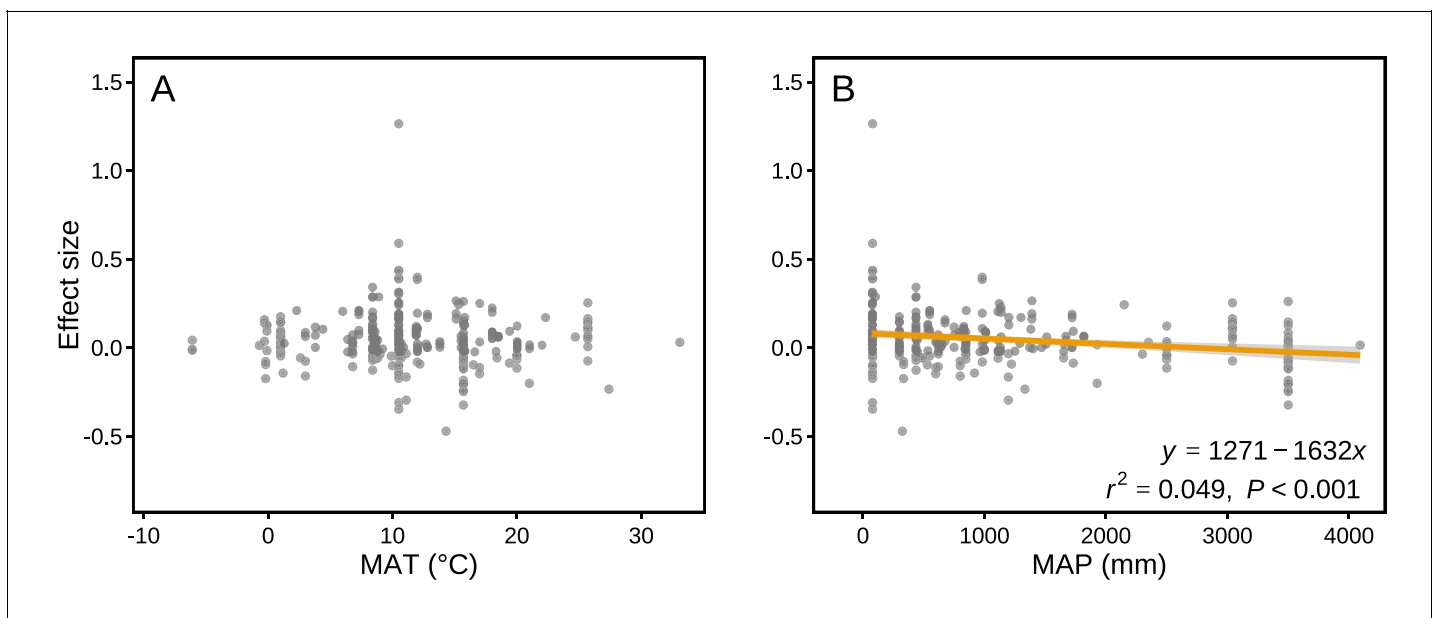

**Figure 6.** Mixture effects on mass loss (effect size) across all studies included in our synthesis as a function of mean annual temperature (MAT; **A**) and mean annual precipitation (MAP; **B**). Data from the final harvest of the decomposition experiment were used to calculate the effect sizes.

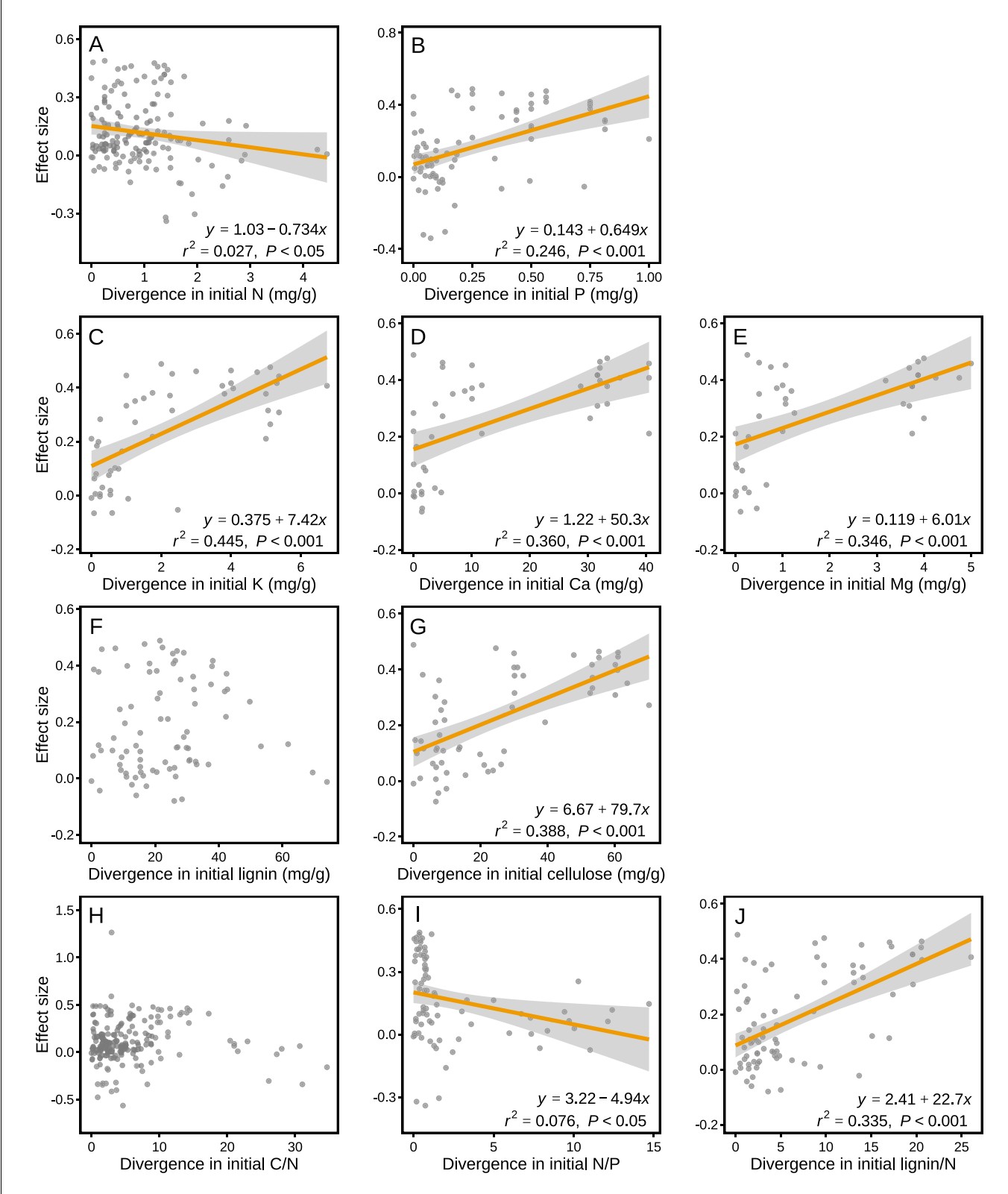

**Figure 7.** Litter mixture effects on mass loss (effect size) as a function of the divergence in initial litter quality among component litter species in a mixture. Trait divergence was calculated following Rao's quadratic entropy (see *Equation 3*) and the Euclidean distance (see *Equation 4*). The mixture effects represent the data for the decomposition process between 10% and 40% of mass loss, during which the strongest non-additive effects were

*Figure 7 continued on next page*

*Figure 7 continued*

observed (see *Figure 3*). The correlations remained significant for Ca ($r^2$ = 0.319, p=0.001) and Mg ($r^2$ = 0.368, p=0.001) even without the data points clustered on the right of the graphs.

## Discussion

### Spatio-temporal patterns of litter mixture effects on C and nutrient dynamics

Our quantitative synthesis of litter mixture effects on decomposition, encompassing most of the Earth's forest biomes, suggests that mixing litter from different plant species generally accelerated litter mass loss and N release compared to what is predicted based on single species decomposition (*Figures 2* and *3*). The predominant synergistic litter mixture effects for both mass loss and N release are in line with a relatively tight coupling of C and N dynamics in terrestrial ecosystems through processes such as biomass production, decomposition, and organic matter storage in soils that all operate within relatively small stoichiometric ranges (*Gruber and Galloway, 2008*). A closer look at each of the five biomes represented in our study, however, reveals a somewhat more hetero-geneous picture. Synergistic litter mixture effects emerged mostly in temperate and subtropical for-ests (*Figure 2*), which may indicate that positive litter mixing effects are restricted to mid-latitude forests, possibly due to climatic interactions. Indeed, a study reported that the presence of non-additive effects was sensitive to temperature, with generally increasing non-additive effects with higher temperatures (*Duan et al., 2013*). However, the predominantly additive effects in the Medi-terranean and the tropical biomes contradict this general temperature sensitivity as did the lack of a relationship between mixture effects and MAT in our analysis (*Figure 6A*), indicating that such sensi-tivity may not hold at the global scale or that other environmental factors interact with temperature. Interestingly, we found a negative, but weak relationship between mixture effects and MAP ($r^2$ = 0.049; *Figure 6B*). Perhaps, high amounts of rainfall increase leaching of labile C and nutrients from decomposing litters (*Pérez-Suárez et al., 2012*), which would restrict non-additive interactions based on transfer and/or complementary use of nutrients and C compounds. It is important to note that the overall pattern is largely driven by the data from temperate forests, which contributed about half of all data points to our meta-analysis. The clear synergistic effects on mass loss and N release, thus, may be a robust characteristic of temperate forests, while forests from other biomes may differ, indicating context-dependent litter mixture effects. However, data limitation for the other forest bio-mes makes it presently difficult to ascertain biome-specific differences in mixture effects, especially for nutrient release with less than 10 studies per biome.

Nitrogen and P are among the most critical nutrients because they have a key role in the metabo-lism of organisms, and thus, they control processes at higher levels of integration, including ecosys-tem productivity (*Elser et al., 2007*; *Vitousek et al., 2010*). Unlike N, the release of P showed no mixture effects irrespective of the identity of biomes (*Figure 2*). Phosphorus is less bound to com-plex C structures and typically released at higher rates than N during litter breakdown (*Manzoni et al., 2010*). In our dataset, an average of 30% of P was released from single species lit-ters compared to 19% of N during the same period of time (*Figure 2—figure supplement 1A*) con-firming a more rapid P than N release. The comparatively rapid P release, especially in subtropical and tropical forests with high rainfall, may thus reduce the potential for positive interactions on P release in litter mixtures. Indeed, the P release rate was about the same in litter mixtures compared to single species treatments, while the N release rate was significantly higher in mixtures compared to single-species treatments. These mixture effects appeared quite insensitive to the number of litter species included in the mixtures, with only additive effects for two- and three-species mixtures (mix-tures with four and more species are strongly underrepresented, *Figure 2—figure supplement 2*; *Supplementary file 2*). The greater N but not P release from litter to soil pools from litter mixtures may indicate a shift in the relative availabilities of these key nutrients with a potential decoupling of N and P cycling with reduced biodiversity. As the absolute demand of decomposer organisms for N is higher than for P, the N:P stoichiometry of available nutrients may be more equilibrated in mixed litter than in single-species litter. Over the longer term, this may cause a shift in the magnitude and direction of ecosystem nutrient limitation in species-poor forests. However, this would need to be

assessed in more detail with additional measurements of N and P availability and microbial responses. Moreover, the results for N and P release need to be interpreted cautiously because of limited data, with less than 10 studies, which were mostly from temperate forests (*Figure 2*).

Interestingly, the overall positive mixing effects on mass loss were significant only during a specific period of the decomposition process. Synergistic mixture effects were particularly clear when mass loss ranged from 10% to 40% and quite weak when mass loss ranged from 40% to 60% (*Figure 3*). Before 10% and after 60% mass loss, the mixture effectswere additive. Synergistic mixture effects, thus, may be particularly important relatively early during the decomposition process, but only after the first few months (10% of mass loss corresponds to 2–6 months globally) during which leaching losses usually dominate litter mass loss. As leaching is mostly a physical process, biological interactions driving potential mixing effects may have a limited effect during this very early stage of decomposition. The synergistic effects decreased gradually over the decomposition process, which can be expected with converging litter quality among litter species during decomposition (*Moore et al., 2006*). However, without continuous measurements of litter quality change through time, this mechanism is difficult to ascertain. Moreover, there is a clear lack of long-term mixed litter decomposition studies. Only three studies measured decomposition to a mass loss of over 80% included in our meta-analysis, which is too few to conclude unambiguously that there are no litter mixture effects in lignin-dominated stages of decomposition. In fact, interactions could even switch to antagonistic mixture effects during the lignin-dominated phase of decomposition (*Preston et al., 2009*), but further studies are required to confirm this. Distinct resource requirements of microbial decomposers (*Allison et al., 2013*; *German et al., 2011*) and the resulting successional changes in microbial community structure and composition during decomposition may also explain the temporal changes in litter mixture effects (*Gessner et al., 2010*; *Wickings et al., 2012*).

Nitrogen release from litter mixtures showed irregular patterns, switching frequently between additive and non-additive effects throughout the decomposition process (*Figure 3*). Nitrogen is often immobilized during the initial stages of decomposition, resulting in a net increase of N. This particular dynamic of N differs fundamentally from mass loss (i.e. mostly C loss) and from P release that is relatively quick and immediate during early decomposition stages. The dynamics of N immobilization and release may have contributed to the shifting mixing effects through time, which was also observed in previous studies (*García-Palacios et al., 2017*; *Schuster and Dukes, 2014*). Nitrogen is also an important driver of the synergistic effect of mixed litter decomposition via N transfer among different litter types (*Lummer et al., 2012*; *Handa et al., 2014*; *Schimel and Hättenschwiler, 2007*). Under N limiting conditions, microbes in decomposing litters can acquire N via multiple pathways, including retention of N released from higher quality litters, reuse of N from microbial necromass, and immobilization of N from the soil pool (*Schimel and Hättenschwiler, 2007*; *Vos et al., 2013*). These alternative ways of N acquisition by microbes in addition to N made available during litter decomposition may result in the greater variability of mixing effects on N release. Overall, however, all litter mixtures as well as single litter species included in our meta-analysis showed net N release (*Figure 2—figure supplement 1*).

## Influences of different components of diversity on decomposition

Litter mixture effects can be driven by different characteristics of mixed litter, such as the number of species, their functional attributes, or the relative contribution of component litter species (*Chapman and Koch, 2007*; *Gessner et al., 2010*; *Hättenschwiler et al., 2005*). Our global analysis clearly shows that there was no relationship between the number of species included in mixtures and the effect size (*Figure 4*). In other words, mixture effects did not increase with species richness. This was also shown in a number of previous case studies (*Barantal et al., 2014*; *Lin and Zeng, 2018*; *Wardle et al., 1997*), which argued that the species composition was far more important than how many species contributed to the mixture. We found that mixtures including four and six species showed additive mixture effects, in contrast to two- and three-species mixtures that both showed synergistic effects. However, because of relatively low numbers of studies including mixtures of six species and the tropical bias in four-species mixtures (*Figure 4—figure supplement 1*), the effects of mixing litter of four and more species need cautious interpretation.

As a rough approximation to evaluate mixture composition effects, we distinguished between deciduous and evergreen plant species, which are typically characterized with a set of contrasting traits along the leaf economics spectrum (*Díaz et al., 2016*). Leaves from evergreen species

commonly have longer lifespans, higher lignin concentrations, and higher ratios of lignin:N and C:N (*Figure 4—figure supplement 2*), and thus a priori slower decomposition rates than leaves from deciduous species (*Aerts, 1995*). We expected that mixing deciduous and evergreen species would result in the strongest mixture effects because the contrasting traits should favor non-additive effects. However, our results only partly confirm this expectation, because mixing deciduous species resulted in similarly strong synergistic effects compared to mixing deciduous with evergreen species (*Figure 4*). On the other hand, when mixtures were composed of only evergreen species the mixture effects were purely additive. The separation into evergreen and deciduous species is a rather coarse approach that neglects a large amount of variability within each of the two functional types that could have contributed to the observed synergistic effects within deciduous species mixtures. Indeed, in a previous large-scale study covering forests from five different biomes, *Handa et al., 2014* showed that interactions between rapidly decomposing and N-fixing species, both deciduous, were driving mixture effects, rather than interactions between deciduous and evergreen. We could not include this finer-grained functional type analysis here, because of the limited number of N-fixing species used in the evaluated studies.

Rather than functional types, differences in specific functional traits among litter species may better quantify functional differences (*García-Palacios et al., 2017*; *Laliberté and Legendre, 2010*). Here, we used the divergence in initial chemical traits among litter species to assess potential relationships between functional divergence and the synergistic mixing effects observed on early-stage decomposition (10% to 40% mass loss). We found a significant relationship between mixing effects and the divergence of a number of initial chemical traits, including P, K, Ca, Mg, cellulose, N:P, and lignin:N (*Figure 7*), but weak or no relationships with the other traits included in the assessment. Similar to the findings of *García-Palacios et al., 2016*, who showed that Mg and Ca played an important role in controlling litter decomposition, these nutrients contributed even more to the litter mixture effects on decomposition in our meta-analysis (*Figure 7C–E*) than the traditionally reported variables (e.g. C, N, and C:N). It has been suggested that Ca and Mg can increase decomposition by stimulating the growth and activity of white rot fungi (*Eriksson et al., 1990*), which may then contribute to N transfer among litter types (*García-Palacios et al., 2016*), potentially explaining the overall strong mixing effects when at least one litter species of high micronutrient concentrations was present in the mixture (thus resulting in high divergence). It should be noted that initial litter quality variables have not been evenly measured across all biomes. For example, about two thirds of the Ca, Mg and K concentration data are from the tropical biome, which may dominate the cation divergence effects on mixed litter decomposition. The study by *Butenschoen et al., 2014* from a montane tropical rainforest contributed many of these data with particularly high divergence in Ca and Mg. The pattern of increasing mixture effects with increasing divergence in Ca and Mg, however, remained robust even when these data were excluded from the analysis (see *Figure 7*). Litter quality parameters other than N, C:N or lignin:N ratios are particularly poorly documented for forests in the boreal, Mediterranean and subtropical biomes, which makes it presently difficult to compare biome-specific relationships or to draw robust conclusions about the generality of the reported relationships between divergence in litter quality and mixture effects on mass loss.

Under natural conditions, different plant species do not typically contribute equally to the leaf litter pool on forest floors (*King, 2002*; *Schuster and Dukes, 2014*). This is only rarely considered explicitly in litter mixture studies. According to the mass ratio hypothesis (*Grime, 1998*), the dominance of certain litter species is expected to prime litter mixture effects, which therefore could differ depending on how evenly the different litter species are represented in a mixture. Our comparison between even and uneven litter mixtures does not suggest fundamental differences with shifts in evenness, as mixtures showed consistently positive effects on mass loss regardless of the relative abundance of litters in a mixture (*Figure 4*), although mixtures with uneven contributions of the different litter species tended to produce slightly stronger synergistic effects. However, overall our results are consistent with a previous study showing that changes in species evenness of litter mixtures had no effect on decomposition (*King, 2002*). Collectively this indicates that mixture effects are not necessarily regulated by the dominant species (*Dj et al., 2013*) as would be predicted by the mass ratio hypothesis. A recent study that explicitly tested for the relative importance of mass ratio and resource complementarity in mixed litter decomposition reported that they both act at the same time to a variable degree, depending on environmental conditions (*García-Palacios et al., 2017*).

## Forest stand structure and mesh size effects

Although natural forests normally possess greater species diversity than planted forests (*Halpern and Spies, 1995*), synergistic interactions emerged in both types of forests, with somewhat stronger effects in plantations on average (*Figure 4*). A more detailed analysis accounting for the distinctive life forms (trees, understory shrubs and herbs) showed that in natural forests, synergistic effects were significant only in mixtures of leaf litter from canopy trees, but not in mixtures where canopy tree and understory species were combined (*Figure 4*). However, we found the opposite for planted forests, where mixtures including both canopy tree and understory species litter showed significant synergistic effects, but not the mixtures composed exclusively of litter from canopy trees (*Figure 4*). We acknowledge that relatively low replication for some of the combinations (e.g. only few studies with litter mixtures including canopy trees and understory herbs for natural forests) makes it difficult to draw general conclusions. Nonetheless, the weaker effects in litter mixtures from canopy trees in planted forests may reflect the artificial canopy composition through the selection of economically valuable tree species. These would not necessarily co-occur naturally and the litter mixtures they produce may not promote interactions among decomposer organisms. On the other hand, the understory community may still be relatively close to that from a more natural forest, which may explain their contribution to synergistic litter mixture effects in plantations. For forest management this means that the understory vegetation has a particularly important role for biogeochemical cycling when forests are managed as plantations.

The litterbag approach to decomposition studies inevitably excludes some of the vast and complex diversity of participating decomposer organisms, depending on the mesh size used. Partial exclusion of the decomposer food web can lead to changes in decomposition (*Bokhorst and Wardle, 2013*; *Bradford et al., 2002*) and the complexity and diversity of decomposer organisms can be an important driver of litter mixture effects (*Barantal et al., 2014*; *Handa et al., 2014*; *Hättenschwiler and Gasser, 2005*; *Vos et al., 2013*). In an attempt to account for the importance of complex decomposer communities, we divided the studies included in our meta-analysis into three groups depending on mesh size (<1 mm, 1 to 2 mm,>2 mm). We could only partially confirm our expectation that mixture effects would increase with increasing mesh size. There was no significant mixture effect for the smallest mesh size, but a clear positive mixture effect in the studies using an intermediate mesh size, that should have allowed access to the majority of mesofauna (*Figure 5*). However, the studies with the largest mesh size, where part of the macrofauna may have contributed to decomposition, and where we expected the strongest non-additive effects, showed no significant mixture effects. This is a surprising result, because soil fauna are important in mediating mixture decomposition as found in several litterbag studies using different mesh sizes and reporting stronger non-additive effects with larger mesh size (*Barantal et al., 2014*; *Martins et al., 2010*; *Schädler and Brandl, 2005*). A difficulty in the interpretation of the mesh size effects is the uneven distribution of different mesh sizes among biomes. For example, almost three quarters of all temperate forest studies used the intermediate mesh size. Temperate forests also contributed the most to the synergistic effects, suggesting that mesh size and biome-specific responses may be confounded. On the other hand, 51% of all observations for larger mesh size are from tropical forests where leaf litter may contain high amounts of polyphenols that may prevent macrofauna feeding (*Hättenschwiler and Vitousek, 2000*). For example, in a neotropical forest fauna was a major driver of mixture effects (*Barantal et al., 2014*). Indeed, we found slightly antagonistic effects for the largest mesh size in tropical forests but not in the other biomes (*Figure 5—figure supplement 1*). It would be interesting to explore the mesh size effect in tropical forests further, as this may depend on plant species and vary among different tropical forests.

Taken together, our quantitative synthesis of litter mixture effects on decomposition in forest ecosystems suggests that mixed litter decomposes more rapidly and releases N at higher rates than single-species litter. These responses are robust for temperate forests, but need confirmation for boreal, Mediterranean, subtropical, and tropical forests that are presently data-limited. Future studies should focus on these presently understudied forest ecosystems by addressing trait divergence in litter mixtures more specifically, which we identified as a major driver of litter mixture effects in early decomposition stages. Based on our results, we conclude that biodiversity loss will modify C and nutrient cycling in forest ecosystems, with the magnitude and direction of changes depending on the complexity of the decomposer communities, litter species properties, and biome. The distinct

responses of N and P release to litter mixing could have particularly far ranging consequences with shifting relative availabilities of these key nutrients potentially decoupling N and P cycling with biodiversity loss. The potential impact of changes in litter diversity on nutrient stoichiometry warrants particular attention in future studies, which ideally would integrate microbial responses for a better understanding of changes in nutrient cycling and driving mechanisms.

## Materials and methods

### Dataset assembly

To develop a comprehensive database, we searched the peer-reviewed articles published prior to 2019 using the ISI Web of Science, Google Scholar, and China National Knowledge Infrastructure. Searches included combinations of the following items: (*leaf* OR *needle* OR *foliage* OR *litter*) AND (*decomposition* OR *decay* OR *breakdown*) AND (*mixing* OR *mixed* OR *mixture* OR *diversity* OR *biodiversity* OR *species composition* OR *species richness* OR *species evenness*). Review or large-scale research articles on a similar topic (*Berglund and Ågren, 2012*; *Gartner and Cardon, 2004*; *Gessner et al., 2010*; *Handa et al., 2014*; *Hooper et al., 2012*) were also used to find studies that were not captured by the searches. Studies that met the following criteria were used in the analysis: (i) the studies focused on decomposition of leaf-litter mixture of any plant life-forms (tree, shrub, and herb) in forest ecosystems, (ii) the decomposition experiments were conducted using litterbags in the field rather than in mesocosms (either under field or laboratory conditions) (iii) the experiments reported mass loss (or mass remaining or decomposition constant), N release or P release of leaf litters decomposed in both isolation (control) and mixture (treatment), and (iv) the means, standard deviations or standard errors, and sample sizes of the selected variables were explicitly reported or could be calculated. Overall, a total of 284 paired observations in 65 published papers covering 63 study sites met these criteria and were included in our study (*Figure 1*; *Supplementary file 1*). These 284 paired observations were from different combinations of 184 species including 149 tree species, 14 shrub species, and 21 herbaceous species (*Supplementary file 3*).

Data were obtained from the text, tables, figures, and appendices of the publications. For figures, data were extracted using GetData Graph Digitizer (version 2.26, http://www.getdata-graph-digitizer.com/). For each selected paper, we recorded the means and stand errors/deviations of mass loss, N and P release. Decomposition constants were transformed into mass loss based on the negative exponential equation used in the original publications. We also obtained chemical traits that were commonly reported, including initial concentrations of N, P, K, Ca, Mg, cellulose, and lignin as well as ratios of C:N, N:P, and lignin:N in leaf litters from original publications. Relevant information was also reported, including geographical location (longitude and latitude), climatic variables (MAT and MAP), and forest type (natural or planted forests). When climate variables were not given, we acquired these data from the WorldClim database (http://www.worldclim.org/) using site location information (latitude and longitude) (*Fick and Hijmans, 2017*) and software ArcGIS 10.3 (http://www.esri.com/software/arcgis/). To assure independence among observations (*Koricheva and Gurevitch, 2014*), only data from the ambient (control) treatment was used in our analysis when studies included treatments such as fertilization, warming, modifications of rainfall, increased $CO_2$, and soil fauna manipulations. Similarly, when a publication reported litter mixtures consisting of the same species, but exposed under multiple conditions, such as different mesh sizes of litterbags (*Schädler and Brandl, 2005*) or exposition and slope (*Mudrick et al., 1994*), we included only one of the observations, choosing the less extreme conditions or the conditions closer to those used in the other studies. For instance, we selected 2 mm rather than 5 mm mesh-size litterbags and slopes of 22% rather than 29%. However, we did not exclude any data from independent studies that met the general criteria when a specific factor (e.g. mesh size) was of interest.

To examine effects of different categorical variables on decomposition of leaf litter mixtures, we grouped the data of mass loss at the final harvest time based on forest biomes (boreal, temperate, Mediterranean, subtropical, and tropical forests), litter species richness (2, 3, 4, and 6 species), litter species evenness (equal ratio vs. non-equal ratio), forest type (natural vs. planted forests), and forest stand structure (represented by the component in the construction of litter mixtures): overstory tree-species composition (tree + tree) vs. vertical stratification among different plant life-forms (tree + shrub, tree + herb, shrub + herb, and tree + shrub + herb). Specifically, a total of 284 paired

observations were included to examine the effects of these categorical variables on mass loss. A total of 240 observations were used to examine effects of plant functional type composition (evergreen + evergreen, deciduous + deciduous, evergreen + deciduous) on mass loss. For each functional type combination, at least one deciduous and/or one evergreen species must have been included in the mixtures. To examine effects of decomposition stage on mixture effects, we included the data of mass loss at all harvest times for each study, resulting in a total of 649 paired observations. Based on the expected values calculated from the mass loss of the individual component species, we created intervals of 10% increments in mass loss up to 70% of mass loss: 0–10%, 10–20%, 20–30%, 30–40%, 40–50%, 50–60%, 60–70%. We did not distinguish further the final interval from 70% to 100%, because only few observations were within this range of mass loss. Due to the limited sample size of N and P measurements, N and P release could only be evaluated for the effects of biome and decomposition stage. Nitrogen and P release were calculated as the concentration at harvest divided by the initial concentration, with values greater than one considered as nutrient immobilization. For comparisons among forest biomes (data at the final harvest time), a total of 69 paired observations from 26 published papers were used for N release and 27 paired observations from 13 published papers were used for P release. A total of 199 paired observations for N release and 73 paired observations for P release were used when testing for the effects of decomposition stage.

## Meta-analysis

Traditional meta-analysis was used to determine the mixing effect of leaf litter on mass loss and release patterns of N and P (*Hedges et al., 1999*). For each study, the effect size was described by the log response ratio (lnRR), which is commonly used in meta-analyses (*Chen et al., 2019*; *Hedges and Olkin, 1985*), and was calculated by *Equation 1*:

$$\text{lnRR} = \ln\left(\frac{X_e}{X_c}\right) = \ln\left(\bar{X}_e\right) - \ln\left(\bar{X}_c\right) \tag{1}$$

where $\bar{X}_e$ and $\bar{X}_c$ are means of observed values (treatment) of decomposition under mixture conditions and expected values (control) calculated from the decay rates of the individual component species, respectively. Its variance was estimated by *Equation 2*:

$$v_{lnRR} = \frac{(s_e)^2}{N_e(s_e)^2} + \frac{(s_c)^2}{N_c(s_c)^2} \tag{2}$$

where $N_e$ and $N_c$ are the sample sizes for the treatment and control groups, respectively; $s_e$ and $s_c$ are the standard deviations for the treatment and control groups, respectively.

The weighted lnRR and 95% bootstrap confidence interval (CI) were calculated using MetaWin software 2.1 (Sinauer Associates, Inc Sunderland, MA, USA) with the random-effect model. If the 95% CIs of the effect sizes for response variables did not overlap with zero (*Hedges et al., 1999*), the differences between observed and expected values (of mass loss, N or P release) were considered significant at p<0.05. To test the influence of categorical factors (forest biomes, decomposition stage, richness, evenness, composition, and forest type and structure) on decomposition, the data were divided into different subgroups. Using an approach similar to one-way weighted ANOVA, we calculated the total variability, which was partitioned into within- and between-group variability (*Hedges et al., 1999*). We then calculated the mean lnRRs and 95% CIs based on the random-effect model as described above. For each categorical factor, the responses of variables were considered significantly (p<0.05) different between observed values and their expected values, if their 95% CIs do not overlap with zero.

To explore influences of climatic factors and initial chemical trait divergence among component litter species in mixtures on decomposition, we used linear regression to test relationships of mixing effect on mass loss with MAT and MAP as well as with divergence in initial concentrations (N, P, K, Ca, Mg, cellulose, and lignin) or key ratios (C:N, N:P, and lignin:N) among component litter species in a mixture. For the evaluation of how trait divergence may explain mixing effects on mass loss, we focused on the period in the decomposition process that showed the strongest non-additive effects (10–40% of mass loss when the effect sizes on mass loss were greater than 5%). For the individual

studies with more than one harvest within this interval of mass loss, we used the average of the effect sizes of mass loss (i.e. only one data point per study). Initial chemical trait divergence was calculated based on Rao's quadratic entropy (*Rao, 1982*):

$$\text{Rao} = \sum_{i=1}^{S-1} \sum_{j=i+1}^{S} d_{ij} p_i p_j \tag{3}$$

where $d_{ij}$ indicates the difference between the $i$-th and $j$-th species and Rao expresses the average difference between two randomly selected individuals with replacements. $S$ is the number of species in the litter mixture. $p_i$ and $p_j$ are the relative abundances of species $i$ and $j$ in the mixture, respectively. The value of $d_{ij}$ is based on Euclidean distance divided by the number of traits, expressed as the mean character difference (*García-Palacios et al., 2017*; *Botta-Dukát, 2005*):

$$d_{ij} = \frac{1}{n} \sum_{k=1}^{n} |X_{ik} - X_{jk}| \tag{4}$$

where $n$ indicates the number of traits considered, $X_{ik}$ represents the value of trait $k$ in species $i$. Due to the highly variable details on litter traits reported in different studies, $n$ equals 1 in our study. The traits included N, P, K, Ca, Mg, cellulose, and lignin as well as C:N, N:P, and lignin:N, for component litter species in a mixture (*Supplementary file 4*). All analyses were conducted in SAS version 9.4 and all figures were plotted in R version 3.6.1.

## Acknowledgements

We thank all the researchers whose data are used in this meta-analysis. We are grateful for the efficient peer-review process overseen by the senior editor Christian Rutz and the reviewing editor David Donoso and for the constructive comments provided by two reviewers Emma Sayer and Nico Eisenhauer on an earlier draft. This research is financially supported by the grants from the National Natural Science Foundation of China (No. 41830646; 31570443) and the National Key Research and Development Program of China (2016YFD0600202). The authors declare no competing interests.

## Additional information

### Funding

| Funder | Grant reference number | Author |
|---|---|---|
| National Natural Science Foundation of China | 41830646 | Shenggong Li |
| National Key Research and Development Program of China | 2016YFD0600202 | Shenggong Li |
| National Natural Science Foundation of China | 31570443 | Shenggong Li |

The funders had no role in study design, data collection and interpretation, or the decision to submit the work for publication.

### Author contributions

Liang Kou, Conceptualization, Writing - original draft; Lei Jiang, Formal analysis, Investigation, Visualization, Writing - review and editing; Stephan Hättenschwiler, Writing - original draft; Miaomiao Zhang, Visualization, Writing - review and editing; Shuli Niu, Xiaoli Fu, Xiaoqin Dai, Han Yan, Writing - review and editing; Shenggong Li, Conceptualization, Funding acquisition, Writing - review and editing; Huimin Wang, Conceptualization, Writing - review and editing

### Author ORCIDs

Liang Kou https://orcid.org/0000-0002-2187-0721

Decision letter and Author response
Decision letter https://doi.org/10.7554/eLife.55813.sa1
Author response https://doi.org/10.7554/eLife.55813.sa2

## Additional files

### Supplementary files

• Supplementary file 1. Location and climatic conditions of studies used in the meta-analysis. Data were extracted from corresponding references or the WorldClim database.

• Supplementary file 2. The relative and absolute N and P release from single litter species treatments and litter mixtures consisting of different numbers of species across all studies included in our meta-analysis.

• Supplementary file 3. Functional type and life form of species included in the meta-analysis.

• Supplementary file 4. Divergence in initial chemical traits among component litter species in 284 mixtures included in this study. Values are calculated based on the *Equation 3* shown in materials and methods. "– "denotes not provided values.

• Transparent reporting form

### Data availability

All data generated or analysed during this study are included in the manuscript and supporting files. Source data are available on Dryad:https://doi.org/10.5061/dryad.nk98sf7qc.

The following dataset was generated:

| Author(s) | Year | Dataset title | Dataset URL | Database and Identifier |
|---|---|---|---|---|
| Kou L, Jiang L, Hättenschwiler S, Zhang M, Niu S, Fu X, Dai X, Yan H, Li S, Wang H | 2020 | Data from: Diversity-decomposition relationships in forests worldwide | https://doi.org/10.5061/dryad.nk98sf7qc | Dryad Digital Repository, 10.5061/dryad.nk98sf7qc |

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
