## [Decision Letter]

**Acceptance summary:**

Leaf litter decomposition is a major ecosystem function, linking plant productivity to carbon stocks in the soil and the atmosphere, and releasing nutrients that shape soil biodiversity. By using a series of meta-analyses, this manuscript evaluated how plant diversity affects nutrient dynamics via leaf litter decomposition, in the major forest biomes across the globe. Importantly, this manuscript found a positive relationship between plant diversity and decomposition rates, but also a decoupling of Nitrogen and Phosphorus release during the process. This manuscript demonstrates the need for large scale synthesis, to understand the links between ecosystem processes and biodiversity.

**Decision letter after peer review:**

Thank you for submitting your article "Robust patterns of diversity-decomposition relationships in forests worldwide" for consideration by *eLife*. Your article has been reviewed by two peer reviewers, and the evaluation has been overseen by a Reviewing Editor and Christian Rutz as the Senior Editor. The following individuals involved in the review of your submission have agreed to reveal their identity: Emma Sayer (Reviewer #1); Nico Eisenhauer (Reviewer #2).

The reviewers have discussed their reviews with one another and the Reviewing Editor has drafted this decision to help you prepare a revised submission.

Summary

The manuscript by Kou and colleagues entitled "Robust patterns of diversity-decomposition relationships in forests worldwide" reports on a global meta-analysis of litter decomposition studies that included litter monocultures and mixtures across forest ecosystems. The authors assembled a large dataset including 270 plant species combinations from a total of 64 field studies and analyzed three main response variables: litter mass loss, N release, and P release. This study aimed at answering the following research questions: (1) Are mixture effects on decomposition non-additive at a global scale, and do they differ among forest biomes? (2) Do mixture effects change during the process of decomposition (i.e. are they stronger in early vs. late decomposition stages)? (3) Are mixture effects related to the divergence of specific litter traits among constituent species? (4) Do mixture effects differ between natural and planted forests?

Essential revisions

1) Overall, the reviewers were supportive of the study and they believe it will be of interest to other researchers working on decomposition processes. However, they feel that the Discussion is rather superficial, and a much stronger case could be made for the wider relevance of this meta-analysis. They can see many opportunities for future studies, and the paper would be of much greater value if these opportunities were properly highlighted.

2) The authors report significantly positive biodiversity effects on decomposition, but also that synergistic effects depend on factors like the stage of decomposition, species richness and functional types of plant litter, functional trait divergence, and the complexity of decomposer communities, and MAP. Nevertheless, the authors conclude that there are "robust patterns of diversity-decomposition relationships" and "consistent litter diversity effects on decomposition in vastly different forest ecosystems across the globe". However, the reviewers do not think that the current main conclusions of this paper are supported by the results. Their main concern is that the authors repeatedly state in the most important sections of the paper (even in the title) that there are globally robust patterns. They are puzzled how the authors came to this conclusion (maybe because of the overall positive effect size of the diversity effect?). They disagree with this conclusion because many deviations from the positive pattern were observed, such as in tropical forests, at late decomposition stages, and at low MAP. This means that basically all of their four initial questions would have to be answered with "yes". Rather than neglecting these differences, the reviewers think that these should be appreciated as exciting research opportunities for future work. They were actually wondering what kinds of results would have led the authors to conclude that they have found context-dependent diversity effects? In fact, the authors introduced several past attempts to synthesize litter diversity effects in meta-studies and highlighted the inconsistency of effects. Rather than showing that these inconsistent effects can be neglected, the present analyses can serve to reconcile inconsistent results by identifying their context-dependency and providing working hypotheses for future research. Careful revisions would need to be made throughout, but the most significant changes are required in title, Abstract, and conclusions.

3) Materials and methods: The reviewers wondered if the arbitrary division of decomposition into 3-month periods makes sense for comparison across biomes. The mixture effects are predicted to have a greater influence at different stages of decomposition, but those stages can occur over distinct timeframes depending on biome/climate and the 3-month division is largely based on temperate studies. Considering the different rates of mass loss during e.g. the first 3-6 months in tropical forest compared to boreal forest would dividing the data into categories based on % mass loss make more sense?

4) Although you generally give the number of studies included in each analysis in the figures, it would be useful to give them in the text at relevant points in the Results, so that the reader has all the relevant information at hand, rather than referring constantly to the figure e.g.: "Specifically, synergistic effects emerged when litter mixtures were composed of two (n = 190) or three species (n = 37) while antagonistic effects predominated when four species (n = 31) were included in a mixture. Additive effect occurred when six species (12) were included in a mixture." This also immediately demonstrates that the two-litter mixtures are most common and that the results for 6 species probably need to be treated with caution.

5) Although it is briefly mentioned at the start of the Discussion that the dataset is dominated by temperate systems, the statement does not fully reflect the actual bias of the data: >1/2 of all datapoints for mass loss and P release and almost 3/4 of the datapoints for N release are from temperate systems. Despite this, the overwhelming influence of temperate data on the "global" patterns for N release are not mentioned. It is also unclear whether there is geographical spread in the analyses of e.g. species numbers and mesh sizes, this would be a discussion point of great interest to other researchers working on decomposition processes.

6) By far the most interesting aspect of the study is the analysis of trait divergence; here, the bias towards certain biomes is discussed, but the discussion of how trait divergence might produce, or influence, mixing effects is rather superficial. Some more information about the links between individual traits and decomposition processes would be very welcome here. For example, the reviewers intrigued by the strong patterns for Ca, because we generally assume that Ca release follows mass loss. In addition, there is a big "gap" in the datapoints in Figure 5 D,E and to a lesser extent in Figure 5I, does the grouping of the data-points correspond to different studies or biomes?

---

## [Author Response]

Essential revisions1) Overall, the reviewers were supportive of the study and they believe it will be of interest to other researchers working on decomposition processes. However, they feel that the Discussion is rather superficial, and a much stronger case could be made for the wider relevance of this meta-analysis. They can see many opportunities for future studies, and the paper would be of much greater value if these opportunities were properly highlighted.

Thank you for your comments. Following your suggestions, we substantially revised the Discussion including more specifically the different points highlighted by our meta-analysis and providing better coverage of opportunities for future studies. For example, we now added more discussion on how trait divergence, especially Ca and Mg, two important nutrients in our study, influence mixing effects. We also added more discussion on the impact of evenness of litter species, which is relevant as under natural conditions, different plant species do not typically contribute equally to the leaf litter pool on forest floors. Considering the contrasting effects of understory species litter in planted and natural forests, we have highlighted the practical significance of our study in regulating forest stand structure in planted forests. Furthermore, we have also revised the discussion regarding mesh size effects on mixture decomposition. Overall, these modifications made the Discussion richer and more balanced, thank you.

2) The authors report significantly positive biodiversity effects on decomposition, but also that synergistic effects depend on factors like the stage of decomposition, species richness and functional types of plant litter, functional trait divergence, and the complexity of decomposer communities, and MAP. Nevertheless, the authors conclude that there are "robust patterns of diversity-decomposition relationships" and "consistent litter diversity effects on decomposition in vastly different forest ecosystems across the globe". However, the reviewers do not think that the current main conclusions of this paper are supported by the results. […] Careful revisions would need to be made throughout, but the most significant changes are required in title, Abstract, and conclusions.

We appreciate this critical look at how we summarized the findings of our meta-analysis. We agree with the reviewers that we initially tried too much to insist on a common pattern across all data combined. We now provide a more balanced presentation of the data, with more in-depth discussion of deviations from the general pattern. Specifically, we revised the title and rephrased relevant sentences throughout the manuscript, especially in the sections of the Abstract, Discussion, and conclusions to better embrace the diversity of responses and to highlight opportunities for future research.

3) Materials and methods: The reviewers wondered if the arbitrary division of decomposition into 3-month periods makes sense for comparison across biomes. The mixture effects are predicted to have a greater influence at different stages of decomposition, but those stages can occur over distinct timeframes depending on biome/climate and the 3-month division is largely based on temperate studies. Considering the different rates of mass loss during e.g. the first 3-6 months in tropical forest compared to boreal forest would dividing the data into categories based on % mass loss make more sense?

This is a pertinent remark, and we initially refrained from this option because the reviewed papers refer the results commonly to time intervals. However, we agree that time intervals do not necessarily reflect decomposition stages. In the revised manuscript, we have reanalyzed the effects of decomposition stage on mixture decomposition based on percentage mass loss (the expected values calculated from the mass loss of the individual component species) instead of time interval. Specifically, we created intervals of 10% increments in mass loss up to 70% of mass loss: 0-10%, 10-20%, 20-30%, 30-40%, 40-50%, 50-60%, 60-70%. We did not distinguish further the final interval from 70-100%, because of only few observations within this range of mass loss. We found that synergistic mixture effects were particularly clear when the mass loss ranged from 10% to 40% and decreased sharply in the range from 40% to 60% of mass loss (Figure 3). Before 10% and after 60%, the mixture effects did not significantly differ from zero. We also reanalyzed the relationships between trait divergence and effect size of mass loss based on these new calculations. The overall pattern is very similar to the previous results based on decomposition time and didn’t change the overall interpretation of Results and conclusions. Please see Results, Discussion and Materials and methods sections of the revised version of the manuscript.

4) Although you generally give the number of studies included in each analysis in the figures, it would be useful to give them in the text at relevant points in the Results, so that the reader has all the relevant information at hand, rather than referring constantly to the figure e.g.: "Specifically, synergistic effects emerged when litter mixtures were composed of two (n = 190) or three species (n = 37) while antagonistic effects predominated when four species (n = 31) were included in a mixture. Additive effect occurred when six species (12) were included in a mixture." This also immediately demonstrates that the two-litter mixtures are most common and that the results for 6 species probably need to be treated with caution.

We now give this information on sample sizes in the text as suggested. We also comment on cautious interpretations when the data base is limited or skewed. Please see paragraph three of the Results.

5) Although it is briefly mentioned at the start of the Discussion that the dataset is dominated by temperate systems, the statement does not fully reflect the actual bias of the data: >1/2 of all datapoints for mass loss and P release and almost 3/4 of the datapoints for N release are from temperate systems. Despite this, the overwhelming influence of temperate data on the "global" patterns for N release are not mentioned. It is also unclear whether there is geographical spread in the analyses of e.g. species numbers and mesh sizes, this would be a discussion point of great interest to other researchers working on decomposition processes.

We appreciate this comment and integrated the dominance of temperate data, as well as bias among biomes (such as the predominant use of intermediate size mesh bags in temperate forests) much better in the text and provided also additional figures to make these points clearer. Please see Discussion paragraphs one and two where we referred to the dominance of data from temperate forests. In the revised manuscript, we added three new figures in the supplementary material (Figure 2—figure supplement 2, Figure 4—figure supplement 1, and Figure 5—figure supplement 1), including the geographical spread in the analyses of species numbers and mesh sizes. However, the sample size is quite small in some of these new figures, which prohibited pushing the differences too much. However, we discuss potential bias and different coverage of data among biomes more explicitly now. Please see Discussion paragraph two and paragraph two of subsection “Forest stand structure and mesh size effects”.

6) By far the most interesting aspect of the study is the analysis of trait divergence; here, the bias towards certain biomes is discussed, but the discussion of how trait divergence might produce, or influence, mixing effects is rather superficial. Some more information about the links between individual traits and decomposition processes would be very welcome here. For example, the reviewers intrigued by the strong patterns for Ca, because we generally assume that Ca release follows mass loss. In addition, there is a big "gap" in the datapoints in Figure 5 D,E and to a lesser extent in Figure 5I, does the grouping of the data-points correspond to different studies or biomes?

Thank you for appreciating the interesting contribution the assessment of trait divergence makes to our manuscript. Following the modified evaluation of the impact of decomposition stage presented in Figure 3, we also changed the analysis of trait divergence by focusing on the period of the strongest non-additive effects during the decomposition process. We improved the Discussion of the observed patterns, especially about Ca and Mg. Indeed, for some litter traits there are distinct data clusters in Figure 5 (Please see Figure 7 now), especially in Figure 7D and 7E. This remained the same with the alternative assessment based on percentage of mass loss instead of time since start of the experiment. We have revisited the dataset and found that one study (Butenschoen et al., 2014) contributed 30 observations (Supplementary file 4) to overall 44 observations from 13 studies spanning five biomes for Ca divergence presented in Figure 7D and to 41 observations from 11 studies spanning four biomes for Mg divergence presented in Figure 7E. Indeed, the data clusters for high Ca and Mg divergence were largely the result of the data from this single study from a tropical montane rainforest. We ran the same analyses without these data and found that the same correlations persisted for Ca (*r*^2^ = 0.319, *P* = 0.001) and Mg (*r*^2^ = 0.368, *P* = 0.001). This suggests that the observed patterns are robust and not driven by one single study or type of ecosystems. We added this more detailed information to the manuscript (see subsection “Influences of different components of diversity on decomposition”).